# DNA Methylation as a Molecular Mechanism of Carcinogenesis in World Trade Center Dust Exposure: Insights from a Structured Literature Review

**DOI:** 10.3390/biom14101302

**Published:** 2024-10-15

**Authors:** Stephanie Tuminello, Nedim Durmus, Matija Snuderl, Yu Chen, Yongzhao Shao, Joan Reibman, Alan A. Arslan, Emanuela Taioli

**Affiliations:** 1Institute for Translational Epidemiology, Icahn School of Medicine at Mount Sinai, New York, NY 10029, USA; stephanie.tuminello@gmail.com; 2Department of Thoracic Surgery, Icahn School of Medicine at Mount Sinai, New York, NY 10029, USA; 3Department of Medicine, NYU Langone Medical Center, New York, NY 10016, USA; 4Department of Pathology, NYU Langone Medical Center, New York, NY 10016, USA; matija.snuderl@nyulangone.org; 5Department of Population Health, NYU Langone Medical Center, New York, NY 10016, USA; 6NYU Perlmutter Comprehensive Cancer Center, New York, NY 10016, USA; 7Division of Environmental Medicine, Department of Medicine, NYU Langone Medical Center, New York, NY 10016, USA; 8Department of Obstetrics and Gynecology, NYU Langone Medical Center, New York, NY 10016, USA

**Keywords:** world trade center, carcinogens, epigenetics, DNA methylation

## Abstract

The collapse of the World Trade Center (WTC) buildings in New York City generated a large plume of dust and smoke. WTC dust contained human carcinogens including metals, asbestos, polycyclic aromatic hydrocarbons (PAHs), persistent organic pollutants (POPs, including polychlorinated biphenyls (PCBs) and dioxins), and benzene. Excess levels of many of these carcinogens have been detected in biological samples of WTC-exposed persons, for whom cancer risk is elevated. As confirmed in this structured literature review (n studies = 80), all carcinogens present in the settled WTC dust (metals, asbestos, benzene, PAHs, POPs) have previously been shown to be associated with DNA methylation dysregulation of key cancer-related genes and pathways. DNA methylation is, therefore, a likely molecular mechanism through which WTC exposures may influence the process of carcinogenesis.

## 1. Background

The World Trade Center (WTC) buildings in New York City exploded and collapsed on 11 September 2001 [1,2]. As each of the 110 concrete floors cascaded down onto the next, a large plume of dust and smoke was generated [1,2]. This WTC dust cloud spread across southern Manhattan and beyond into areas of Brooklyn [3]. Aerosolized dust settled in both outdoor and indoor locations. Outdoor dust settled on streets, sidewalks, and buildings, with dust from external ledges and automobiles used for subsequent chemical and physical analyses. Settled outdoor WTC dust was unique in terms of particle size distribution. Out of the total mass of settled dust, >90% of the particles were between 2.5 and 100 μm [2,3,4,5]. Those exposed to the initial dust clouds created as the buildings collapsed (acute exposure) were likely exposed to a higher proportion of smaller (<2.5 μm) airborne particles [2]. WTC dust also settled indoors, having been blown through shattered or open windows, or through air ventilation systems. Indoor dust, while physically and chemically similar to the outdoor settled dust, was also found to contain a greater percentage of more easily inhalable particles (<53 μm) [4]. In some indoor and outdoor areas, settled dust was inches thick [2,3,4]. With a mass median aerodynamic diameter of just 23 μm, there was also a high potential for airborne resuspension [2,6].

Acute exposures to this WTC dust cloud occurred in emergency rescue and recovery workers (responders), and in those working, living, attending school, or just passing by in the area (survivors) on 11 September 2001 [7]. In these individuals, there was potential for acute dust inhalation (being caught in the dust cloud), as well as ingestion of the dust and body surface exposure. The fine-particle nature of the WTC dust allowed for it to penetrate through the lung conductive airways to reach the gas exchange airways in the deep lung of those who were exposed [2]. In addition to acute exposures on 11 September 2001, chronic exposures to the dust or fumes/smoke occurred in responders and survivors in the subsequent months [7]. These exposures resulted from the fires that persisted through December 2001, from dust during recovery and clean-up work, and from resuspended dust in the streets, incompletely or poorly cleaned indoor areas, and in buildings ventilation systems [7].

Many of the components of WTC dust are known or suspected human carcinogens, and overall cancer rates of WTC responders are 6–14% higher than background rates [3,4,8,9,10]. Specifically, responders suffer from excess prostate, thyroid, skin, and tonsil cancer [8,9,10,11,12,13,14]. The potential exists that routine screening administered to WTC-exposed persons introduces detection bias and artificially elevates cancer diagnosis rates, at least for WTC thyroid and prostate cancer cases [15]. Yet, there is evidence of a dose–response relationship between having arrived earlier to the disaster sites and prostate, thyroid, skin, and tonsil cancer incidence [11,16]. Greater WTC exposure has also been associated with more advanced clinical stage among WTC responders with prostate cancer [17]. Cancers in the nonresponder population (survivors) are commonly observed but remain understudied [18]. More than 6000 cancer patients, including 1300 breast cancer patients, have been diagnosed among WTC survivors at the WTC Environmental Health Center (WTC EHC) as of 1 November 2023 [14,18,19]. While studies lack statistical power to detect significant association of WTC exposures with breast cancer as women were under represented in WTC responders, initial studies among survivors suggest that WTC-associated breast tumors appear more likely to be poorly differentiated, and of aggressive molecular subtypes compared to the general population [20]. WTC-associated cancers may, therefore, be more aggressive compared to cancers in the general population [17,21,22]. Given the latency period for cancer development, and the aging population of WTC-exposed individuals, rates of WTC-associated cancer are expected to rise. Thus, a better understanding of the underlying pathways between WTC exposures and cancer development is urgently needed.

The mechanisms by which exposure to the toxicants in the WTC dust has affected the health of WTC-exposed persons are not fully understood. Many of the compounds identified in the WTC dust have previously been shown to modify epigenetic status. Epigenetic mechanisms act as the interface between the genome and the environment. DNA methylation changes are chemical modifications to the genome that occur in response to exogenous exposures, which influence gene regulation and expression [23]. More specifically, DNA methylation occurs when there is the addition of a methyl group to the cysteine base at a cytosine–phosphate–guanine (CpG) site [24,25]. When DNA methylation occurs at the gene promoter regions, it often results in the silencing of gene expression [25]. DNA methylation is dynamic and actively maintained throughout the genome, but can be dysregulated in cancer [24]. Aberrant DNA methylation is a hallmark of cancer development and progression [26,27]. Because (1) cancer cells are characterized by aberrant DNA methylation patterns, and (2) tumor suppressor genes, specifically, have been found to be targets to DNA hypermethylation early on in the carcinogenic process [28], we hypothesize that DNA methylation is the form of epigenetic modification most pertinent to WTC-exposure-associated carcinogenesis. DNA methylation has, in fact, been described as “at the crossroads of gene and environment interactions” [29]. Moreover, our group’s previous findings from case–control epigenome-wide association studies of WTC exposure have shown DNA methylation changes in important cancer-related genes and pathways. For instance, enrichment in differentially methylated genes belonging to the MAPK signaling, mTOR signaling, base excision repair, epithelial–mesenchymal transition, and hypoxia pathways, among others, occurred for WTC-exposed persons [30].

Here, we sought to (1) describe the reported carcinogens contained within the WTC dust and the detection of these carcinogens among WTC-exposed individuals, and (2) conduct a structured review of the literature on DNA methylation alterations associated with exposure to carcinogens present in the WTC dust, to identify potential biological avenues whereby acute and chronic WTC dust and fumes exposures may be contributing towards carcinogenesis. 

## 2. Carcinogens Present in the WTC Dust and Studies of WTC-Exposed Populations

The International Agency for Cancer Research (IARC) classifies group 1 carcinogens as substances with “sufficient evidence of carcinogenicity in humans”. Multiple IARC Group 1 carcinogenic agents were present in the WTC settled aerosolized dust, as measured in both outdoor and indoor samples. These agents include metals (arsenic (As), beryllium (Be), cadmium (Cd), chromium (Cr), and nickel (Ni)); asbestos; polycyclic aromatic hydrocarbons (PAHs); persistent organic pollutants (POPs), including polychlorinated biphenyls (PCBs) and dioxins; and volatile organic compounds (VOCs), including benzene [3,4,31] (Table 1). Other, noncarcinogenic chemicals and toxicants were also detected in the outdoor and indoor dust samples, as reported by the chemical analyses of Lioy et al. and Yiin et al. [3,4].

Excess levels of many of these carcinogens have been detected in biological samples of WTC responders and survivors. An analysis of blood and urine specimens from 321 WTC-exposed firefighters versus 47 unexposed controls detected increased fire-retardant chemicals, and cadmium, PAH, and dioxin, among WTC responder firefighters [32]. This study took place within the first months after the WTC tower collapse. Similarly, in blood plasma samples collected retrospectively from New York State employees and National Guard personnel (n = 43) assigned to work in the vicinity of WTC during the week after the buildings’ collapses, elevated levels of dioxin-like polychlorinated dibenzofurans (PCDFs) were observed among those with greater dust and smoke exposure [33]. Pregnant women survivors have also been observed to have high levels of carcinogenic exposures. Among 187 pregnant persons who were at or near the WTC disaster site, the highest levels of PAH-DNA adducts were seen in blood collected immediately after 11 September 2001, suggesting bystander exposure [34]. A nonsignificant positive association between elevated polychlorinated biphenyls (PCBs) levels and greater WTC exposure was also observed [34]. PAH-DNA adducts in maternal (n = 170) and cord blood (n = 203) were found to be at the highest levels among mothers that resided in close geographical distance from the WTC disaster site [35], and were elevated compared to those mothers living in northern Manhattan [36]. In another study, 210 cord blood specimens from mothers who delivered in hospitals in lower Manhattan were investigated for flame-retardant chemicals (polybrominated diphenyl ethers (PBDEs)). Although cord plasma levels of PBDEs were not significantly associated with distance from the WTC site, women who were in the second half of their pregnancy on 11 September 2001 had children with higher cord concentrations of PBDEs, suggesting some relationship between gestational age and in utero WTC dust exposure [37]. A different study of 108 mothers from a Columbia birth cohort did find that geographical exposure via proximity to the WTC was associated with higher dioxin exposure, which may have influenced birth outcomes [38]. Similarly, WTC-exposed children may have been especially vulnerable. Fire-suppressant materials, specifically perfluoroalkyl substances (PFASs), were found to be elevated in serum from 123 children who were ≤8 years of age on 11 September 2001, compared to 185 sociodemographically matched controls [39]. Other studies have corroborated this result [40]. Adolescents exposed to the WTC disaster also had measurable exposure to WTC dust. WTC-exposed youths who lived, attended school, or were present in lower Manhattan compared to unexposed youths frequency-matched on age, sex, race, ethnicity, and income had significantly elevated serum dioxin levels, even years after 11 September 2001 [41]. Among WTC-exposed community members decedents, acute exposure to the WTC dust cloud was significantly associated with increased levels of chromium (Cr) and cadmium (Cd) as measured in lung tissue [42] (Table 1).

**Table 1 biomolecules-14-01302-t001:** Summary of IARC Group 1 carcinogens in the WTC outdoor and indoor settled dust.

Carcinogen Group	Carcinogens	Abundance in Outdoor WTC Dust [3]	Abundance in Indoor WTC Dust [4]	Carcinogen Elevated among WTC-Exposed
Metals and Metalloids [1,4,5,31,43]	Arsenic (As) Beryllium (Be) Cadmium (Cd) Nickel (Ni) Chromium (Cr)	~2623 ng/g dry weight ~3228 ng/g dry weight ~7203 ng/g dry weight ~43,490 ng/g dry weight ~165,367 ng/g dry weight	~3123 ng/g dry weight ~1805 ng/g dry weight ~3700 ng/g dry weight --- ~69,325 ng/g dry weight	Survivors; lung tissue [42]
Chrysotile asbestos [1,31,43]	---	0.8–3% by mass	<1% volume	n/a
Polycyclic aromatic hydrocarbons (PAHs) [31]	40 different compounds including: Benzo[a]pyrene	Total PAHs: ~325,833 ng/g ~18,133 ng/g dry weight	1000–5000 ng/g dry weight per individual compound	Firefighter responders; blood and urine [32] Pregnant women; maternal and cord blood [34,35,36]
Persistent Organic Pollutants (POPs):	Polychlorinated biphenyls (PCBs): [1,31] 68 different congeners Polychlorinated dibenzodioxins (PCDDs), or simply dioxins [1,4,31]: 7 types, including: 2,3,7,8-TCDD	~1306 ng/g dry weight Total dioxins: ~100 ng/kg ~6.45 ng/kg	--- ~100 ng/kg ---	New York State employees and National Guard personnel; blood [33] Pregnant women; blood and urine [34,38] Adolescents; blood [41]
Hydrocarbon (HC) and Volatile Organic Compounds (VOCs) [1,4,31,44]	Benzene	>5000 ng/g dry weight per individual compound	<level of quantitation	n/a

## 3. DNA Methylation Associated with WTC Carcinogens: A Structured Literature Review

The carcinogens present in the WTC dust are also known epigenetic modifiers. A structured literature review on the effects of these agents on DNA methylation status was completed through a PubMed search strategy as of 13 July 2023 (the search strategy is detailed in Appendix A). The literature inclusion criteria were as follows: (1) The study examined human participants. (2) Study participants were exposed to one or more of the following WTC-associated carcinogens: (arsenic (As), beryllium (Be), cadmium (Cd), chromium (Cr), and nickel (Ni)); asbestos; polycyclic aromatic hydrocarbons (PAHs); persistent organic pollutants (POPs), including polychlorinated biphenyls (PCBs) and dioxins; and volatile organic compounds (VOCs), including benzene. (3) Exposure occurred in the postnatal period. (4) Either global or cancer-associated, CpG site-specific DNA methylation was assessed post-exposure in blood or tissue specimens. (5) Studies were reported in the English language. Case studies and case series were excluded. An additional fourteen relevant review articles were assessed for identification of additional relevant studies [23,24,45,46,47,48,49,50,51,52,53,54,55,56]. For efficient article screening, Covidence systematic review software (Veritas Health Innovation, Melbourne, Australia) was used, which is available at www.covidence.org (accessed on 13 July 2023). A data abstraction template was developed before screening. It included the following information: study author and title, study design (including participants and exposure assessment), outcome measurement (sample type for DNA extraction, DNA methylation platform, and statistical analysis methods), and results (global DNA methylation pattern or statistically significant, site-specific methylation changes). Differentially methylated genes were determined to be cancer-related, and classified as either tumor suppressors or oncogenes, by querying the NCG7.1 Network of Cancer Genes [57].

A total of 1146 studies were identified and screened for relevance, of which 977 did not meet the inclusion criteria for this review. The full texts of the remaining 169 articles were reviewed for inclusion. In total, 62 studies were found to meet inclusion criteria, with an additional 18 eligible articles identified from previously published review articles [23,24,45,46,47,48,49,50,51,52,53,54,55,56], resulting in the inclusion of 80 articles (see Figure 1 for Preferred Reporting Items for Systematic Reviews and Meta-analyses (PRISMA) guidelines). A more detailed description of these 80 studies can be found in the Appendix A. 

The number of studies identified for each carcinogen category was as follows: metals: 38 (As: 15, Be: 1, Cd: 7, Cr: 5, Ni: 1, combination of metals: 9), asbestos: 4, benzene: 15, PAHs: 15, POPs: 8, PCBs: 7, PCBs + dioxins: 1. 

All the carcinogens present in the settled WTC dust (metals, asbestos, benzene, PAHs, POPs) were associated with DNA methylation dysregulation, specifically of tumor suppressor and/or oncogenes. Results are described fully in Table 2 and summarized below. Additional details about each study, including direction of DNA methylation (increase or decrease) per gene for each carcinogen, are provided in Appendix A. 

### 3.1. Arsenic (As)

Arsenic was the most investigated metal (n = 23 studies (n = 15 alone, n = 8 in combination with other metals)) in the literature [58,59,60,61,62,63,64,65,66,67,68,69,70,71,72,73,74,75,76,77,78,79,80].

Of the 14 studies reporting on global patterns of DNA methylation, 8 reported increased global DNA methylation after arsenic exposure. 

Differential DNA methylation of nine tumor suppressor genes and three oncogenes was observed.

### 3.2. Beryllium (Be)

One study reported on beryllium exposure and DNA methylation [81].

This study found decreased global DNA methylation and differential methylation of one oncogene (*CXCR4*).

### 3.3. Cadmium (Cd)

Cadmium exposure was assessed by 12 studies (n = 7 alone, n = 5 in combination with other metals) [65,70,74,79,80,82,83,84,85,86,87,88].

Seven studies assessed global DNA methylation and reported mixed results. 

Differential DNA methylation of 14 tumor suppressor genes and 4 oncogenes was observed.

### 3.4. Chromium (Cr)

DNA methylation post-chromium exposure was investigated in 7 studies (n = 5 alone, n = 2 in combination with other metals) [65,89,90,91,92,93,94].

Two out of three studies reported chromium-associated increased global DNA methylation. 

Differential DNA methylation of four tumor suppressors was observed.

### 3.5. Nickel (Ni)

Nickel-associated DNA methylation was assessed in 5 studies (n = 1 alone, 4 in combination with other metals) [68,72,86,89,95].

Two out of three studies reported nickel-associated increased global DNA methylation.

Differential methylation of four tumor suppressor genes and one oncogene was observed. 

### 3.6. Asbestos

Just four studies reported on DNA methylation changes after asbestos exposure [96,97,98,99].

One study reported decreased global DNA methylation. 

Differential DNA methylation of three tumor suppressor genes was observed. 

### 3.7. Benzene

Benzene exposure and associated DNA methylation changes was explored in 15 studies [100,101,102,103,104,105,106,107,108,109,110,111,112,113,114].

Of the nine studies reporting on global patterns of DNA methylation, seven reported decreased global DNA methylation.

Differential DNA methylation 10 tumor suppressors and 8 oncogenes was observed. 

### 3.8. Polycyclic Aromatic Hydrocarbons (PAHs)

Fifteen studies were identified that explored the impact of PAH exposure on DNA methylation [115,116,117,118,119,120,121,122,123,124,125,126,127,128,129].

Seven studies assessed global DNA methylation and reported mixed results. 

Differential DNA methylation of nine tumor suppressors and two oncogenes were observed. 

### 3.9. Persistent Organic Pollutants (POPs)

Eight studies reported on the DNA methylation consequences of POP exposure. More specifically, seven reported on PCB exposure [130,132,133,134,135,136,137] and one reported on both PCBs and dioxins [131].

Of the six studies reporting on global patterns of DNA methylation, four reported increased global DNA methylation.

Differential DNA methylation of five tumor suppressors and six oncogenes was observed. 

## 4. Discussion

While the risk for cancer among WTC-exposed responders is now well known, it is still not understood which specific components of the WTC dust may be driving the cancer risk, or what biological mechanisms may be at play. As reviewed here, WTC dust contained known carcinogenic materials, and some of these carcinogens have been found to be elevated in the biological specimens of WTC-exposed responders and survivors. Additionally, as confirmed in this literature review, all carcinogens present in the settled WTC dust (metals, asbestos, benzene, PAHs, POPs) are associated with DNA methylation dysregulation. Exposure to any of these alone, or in combination, could feasibly contribute to carcinogenesis among WTC-exposed responders and survivors by disrupting genetic regulation in key cancer-related genes and pathways.

This work is in keeping with our prior findings that WTC exposure is linked to increased global DNA methylation. Across epigenome-wide association studies (EWASs) of cancer-free WTC survivors, WTC-exposed survivors with breast cancer, and WTC-exposed responders with prostate cancer [30,138,139,140], we have observed increased global DNA methylation and statistically significant alterations in the DNA methylation levels of cancer genes and pathways among WTC-exposed persons [30]. This was observed in both blood and tissue. For instance, tumor suppressor *BRCA1* has previously been observed to be hypermethylated among WTC-exposed persons, and its hypermethylation is associated with PAH exposure as reviewed here. Traditionally, gene hypermethylation is associated with silencing gene expression [141], suggesting that important tumor suppressor genes could have been “turned off” after WTC exposure. Preliminarily, it appears that exposure doses of carcinogens from WTC dust may be high enough to induce DNA methylation changes, but this needs to be more fully explored. It should be acknowledged that this literature review only summarizes potential biological avenues whereby WTC dust may be contributing towards carcinogenesis, without providing direct evidence that exposure of individuals to WTC dust resulted in the DNA methylation change mentioned here. Additional direct evidence from larger EWAS and animal studies is warranted to test these hypothesized associations.

It is true that epigenetic alterations are generally less stable as genetic alterations; nevertheless, they are retained long term, through multiple rounds of cellular division [142,143]. DNA methylation of the promotor regions of tumor suppressors, such as genes involved in DNA repair, have been shown to be stable and permeant in tumor tissues [144]. Moreover, it is now well established that genetic and epigenetic mechanisms are related events in cancer [145]. Simplified, epigenetic alterations can lead to genetic mutations, and in turn genetic mutations in epigenetic regulators can lead to a furthered altered epigenome; this co-accumulation of genetic and epigenetic alterations is associated with increased cancer risk [145]. Thus, DNA methylation changes from environmental exposures are likely to be persistent throughout the carcinogenesis process.

This review begins to address the question of whether specific carcinogens in the WTC dust could have had greater or lesser impact on a biological level. Metals in the WTC dust appear to play a significant role. Among the different carcinogen types, metals exposure and associated DNA methylation alterations have been the most studied (n = 38 studies). Arsenic is the most studied metal type in terms of DNA methylation changes. Metals are not known to be highly mutagenic; instead, epigenetic mechanisms underlie their carcinogenic potential [46]. As summarized here, multiple tumor suppressors and oncogenes are likely dysregulated in response to metal exposure. Exposure to metals may be associated with increased global DNA methylation.

The link between asbestos exposure and DNA methylation is less well studied (n = 4 studies), but tumor suppressor dysregulation has still been observed among asbestos-exposed individuals. Studies of the epigenetic consequences of benzene exposure are more numerous (n = 15), and, again, DNA methylation alterations in cancer-related genes are consistently reported. Benzene exposure may also be associated with decreased global DNA methylation, although previous studies’ findings vary. PAH (n = 15 studies) and POP (n = 8 studies) exposures are genotoxic [49], but, as reviewed here, DNA methylation changes associated with PAH and POP exposures are commonly observed. Both PAH and POP exposures are associated with disrupted epigenetic regulation of tumor suppressors and oncogenes.

Dysregulation of certain cancer genes appears to be especially relevant. For instance, dysregulation of the *WNT* tumor suppressor gene is associated with As, Cr, Ni, and PAH exposure. This gene encodes for the Adenomatous Polyposis Coli (APC) protein, which regulates Wnt signaling, but is also critical for cytoskeletal structure [146]. Immune system and cell mobility dysfunction are both commonly observed among WTC responders and survivors [30]. Likewise, in the literature, *p16^INK4A^ (CDKN2A)* DNA methylation alteration is associated with all known WTC carcinogens except dioxins. *CDKN2A* is important for cell cycle control, but also for lipid metabolism, another commonly dysregulated set of pathways among WTC-exposed persons [30].

Notable limitations for this work warrant discussion. Firstly, while carcinogenic chemicals were identified in the settled dust, there are incomplete data on aerosolized dust samples and a lot of variability in sample content depending on the sampling site. Moreover, systemic and complete measures of biologic markers of these chemicals is lacking. Many were not measured or were not measurable; some measures are available only for discrete populations. The time points for measurements were not consistent, or measures were delayed. For example, some toxicants, like PFAS, dioxins, and furans, were not initially measured either in the indoor or outdoor settled dust, as their harm for human beings was not appreciated twenty years ago, but their detection has been documented in WTC-exposed children and other community members [36,37,38]. Moreover, some carcinogens, like asbestos and benzene, have yet to be directly measured in the blood, urine, or tissues of WTC-exposed persons, although their detection has been documented in individual cases. Notably, only a single study has measured WTC dust components directly in the tissue of exposed persons [42]. These represent gaps that future research should address as well as reinforce the importance of documenting exposures and obtaining blood samples in other environmental disasters. We would like to note, additionally, that this review only covers DNA methylation alterations, while carcinogens have also been linked to other types of epigenetic mechanisms, such as histone modifications [147]. Moreover, beyond exposure to the dust itself, other elements of the WTC disaster event, such as the stress and trauma, may have caused epigenetic modifications [148]. Future studies should explore these issues to develop a broader understanding of WTC exposure and cancer causality.

Notable gaps also exist in the available literature. Certain WTC-associated carcinogens, such as asbestos, remain understudied in terms of their impacts on the epigenome. Moreover, only a fraction of the published literature reported on global DNA methylation patterns, which, given its biological importance and relevance to carcinogenesis, warrants greater research efforts in the future. While all cancer types exhibit DNA methylation changes, it has been observed that certain epigenetic changes are tissue-specific [26]. The literature utilized in this review was not limited to tissue-based studies, but also included studies whereby DNA was extracted from blood and other sample types. Thus, it is possible that certain tissue-specific DNA methylation changes were not captured.

In summary, there is strong evidence that DNA methylation is a molecular mechanism through which WTC exposures may influence the process of carcinogenesis. Evidence-based DNA methylation changes associated with chemicals present in the WTC dust should be further assessed among WTC-exposed responders and survivors. Correlation of these epigenetic changes with gene expression patterns, and assessment of the heritability of such epigenetic marks, will require further exploration. This could lead to the development of novel biomarkers for cancer detection and prognosis among WTC-exposed populations.

## Figures and Tables

**Figure 1 biomolecules-14-01302-f001:**
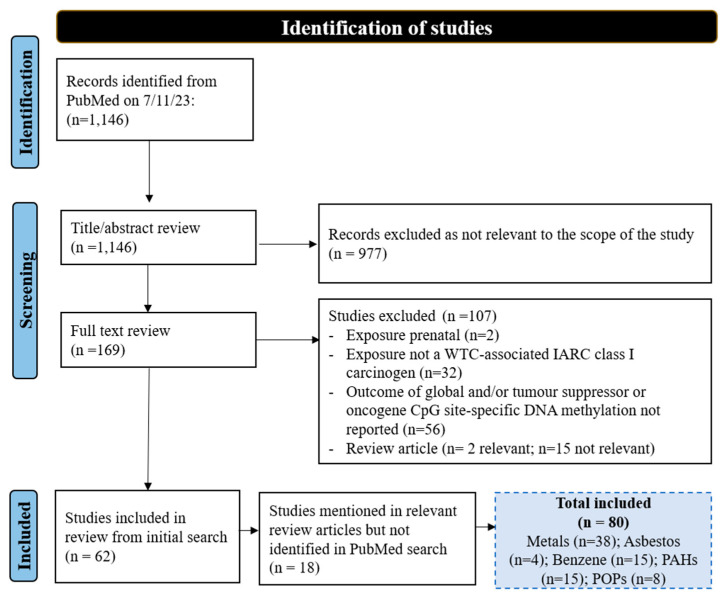
Selection of relevant literature.

**Table 2 biomolecules-14-01302-t002:** Summary of carcinogenic metals, asbestos, benzene, POPs, PAHs, and associated global and/or cancer-specific differential DNA methylation.

Carcinogen	Global DNA Methylation Pattern	Differentially Methylated Genes	
		Tumor Suppressors	Oncogenes
Arsenic (As) * (n = 23 studies [n = 15 alone, n = 8 in combination with other metals])	Increased: 8 studies [58,59,60,61,62,63,64,65] Decreased: 3 studies [66,67,68] No association: 3 studies [69,70,71]	*APC* [72], *CREBBP* [58], *DAPK* [73], *MLH1* [74], *p16^INK4A^ (CDKN2A)* [72,74,75,76,77], *PRDM2* [63,78], *RASSF1A* [72], *SUFU* [63], *TP53* [72,75]	*ACKR3* [79], *CTNNA2* [64], *GNAS* [80]
Beryllium (Be) (n = 1, alone)	Decreased: 1 study [81]	*---*	*CXCR4* [81]
Cadmium (Cd) (n = 12 studies [n = 7 alone, n = 5 in combination with other metals])	Increased: 2 studies [65,82] Decreased: 3 studies [83,84,85] No association: 2 studies [70,86]	*ARID1B* [80], *ARHGEF10* [80], *BRCA2* [87], *CBFB* [87], *CD79B* [87], *ETV6* [87], *GATA3* [87], *IGF2BP2* [80], *LARP4B* [80], *MLH1* [74], *RET* [87], *SMAD3* [80], *TGFBR2* [87], *ZBTB16* [80]	*DNMT1* [74], *NFE2L2* [87], *RARA* [87,88], *RET* [87]
Chromium (Cr) (n = 7 studies [n = 5 alone, n = 2 in combination with other metals])	Increased: 2 studies [65,89] Decreased: 1 study [90]	*APC* [91], *p16^INK4A^ (CDKN2A)* [92], *MGMT* [91,93], *MLH1* [94]	*---*
Nickel (Ni) (n = 5 studies [n = 1 alone, n = 4 in combination with other metals])	Increased: 2 studies [68,89] No association: 1 study [71]	*APC* [72], *p16^INK4A^ (CDKN2A)* [72], *RASSF1A* [72], *TP53* [72]	*p15 (ABL1)* [95]
Asbestos (n = 4)	Decreased: 1 study [96]	*DFNA5 (GSDME)* [97], *EDAR* [97], *p16^INK4A^ (CDKN2A)* [98,99]	*---*
Benzene (n = 15)	Increased: 2 studies [100,101] Decreased: 7 studies [102,103,104,105,106,107]	*CDH1* [100], *ERCC3* [108,109], *FAS* [100], *FAT1* [100], *MGMT* [104,110], *MLH1(hMLH)* [104], *p16^INK4A^ (CDKN2A)* [111], *NRG1* [100], *SMAD3* [100]	*ALK* [100], *CSF1R* [100], *CSF3R* [100], *GNAS* [100], *p15 (ABL1)* [102,112,113], *NSD2* [100], *PRDM16* [114], *STAT3* [101]
PAHs (n = 15)	Increased: 3 studies [115,116,117] Decreased: 2 studies [118,119] No association: 2 studies [120,121]	*APC* [115,118], *BRCA1* [120], *CDH1* [120], *DAPK* [120], *FAT1* [122], *p16^INK4A^ (CDKN2A)* [118,121,123,124,125,126], *MLH1* [118], *MGMT* [121,127,128], *TP53* [117,129]	*CCND2* [120], *ESR1* [120]
PCBs/Dioxins ^ (n = 8)	Increased: 2 studies [130,131] Decreased: 4 studies [132,133,134,135]	*ARHGEF12* [136], *FBXW7* [136], *IGF2BP2* [130], *MGMT* [137], *PTEN* [136]	*ABL2* [130], *ARAF* [130], *BCL11A* [136], *LCK* [130], *MITF* [136], *REL* [136]

* If metals were assessed in combination, differentially methylated patterns and genes are reported as modified by each individual carcinogen. ^ These studies report on PCBs only, except for Lind et al., 2013, which also reports on dioxin exposure.

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
