# Peer review of "DNA Methylation as a Molecular Mechanism of Carcinogenesis in World Trade Center Dust Exposure: Insights from a Structured Literature Review"

_biomolecules, 2024, doi:10.3390/biom14101302_

Round 1

Reviewer 1 Report (Previous Reviewer 2)

Comments and Suggestions for Authors

The reviewer did not know what exact modifications the authors have made to improve the manuscript because they always highlighted the whole paragraph in the change tracking file. In addition, epigenetic alterations are generally not as stable as genetic alterations. It will be helpful for readers if they could mention studies on the stability of epigenetic alterations of tumor-related genes (such as CDKN2A/p16, MLH1, LINE-1, and DAPK) during carcinogenesis to indicate that epigenetic alterations were not only transiently induced by the exposure of carcinogens from WTC dusts but also persistently maintained in target cells (such as transformed cells in precancerous or cancerous lesions)

Author Response

The reviewer did not know what exact modifications the authors have made to improve the manuscript because they always highlighted the whole paragraph in the change tracking file. In addition, epigenetic alterations are generally not as stable as genetic alterations. It will be helpful for readers if they could mention studies on the stability of epigenetic alterations of tumor-related genes (such as CDKN2A/p16, MLH1, LINE-1, and DAPK) during carcinogenesis to indicate that epigenetic alterations were not only transiently induced by the exposure of carcinogens from WTC dusts but also persistently maintained in target cells (such as transformed cells in precancerous or cancerous lesions)

We apologize for any confusion. The entirety of the paragraph was highlighted as it was newly added to provide additional context about our previous epigenome-wide association studies (EWAS) of WTC exposure, which demonstrated global changes in DNA methylation profiles, as well as specific changes in cancer-related genes and pathways associated.

We thank reviewer 1 for their comment regarding the long-term stability of DNA methylation changes during cancer development. While it is true that DNA methylation is less stable than genetic changes, it has been established that DNA methylation is a stable, rich source of information on tumor biology. Towards this point, we have expanded our discussion, “It is true that epigenetic alterations are generally less stable as genetic alterations; never the less they are retained long-term, through multiple rounds of cellular division.140,141 DNA methylation of the promotor regions of tumor suppressors, such as genes involved in DNA repair, have been shown to be stable and permeant in tumor tissues.142 Moreover, it is now well established that genetic and epigenetic mechanisms are related events in cancer.143 Simplified, epigenetic alterations can lead to genetic mutations, and in turn genetic mutations in epigenetic regulators can lead to a furthered altered epigenome; this co-accumulation of genetic and epigenetic alterations is associated with increased cancer risk.143 Thus DNA methylation changes from environmental exposures are likely to be persistent throughout the carcinogenesis process.” (Lines 276-285)

Reviewer 2 Report (Previous Reviewer 1)

Comments and Suggestions for Authors

This structured review discusses DNA methylation as a potential mechanism by which specific chemicals from WTC dust act as carcinogens. The authors searched Pubmed for studies that investigated DNA methylation changes in humans exposed to carcinogens that had been detected in individuals exposed to WTC dust. Eighty studies were included in their analysis. These studies indicated that various carcinogens altered DNA methylation at specific genomic sites. However, there is no evidence that the levels or exposures of individuals exposed to WTC dust result in any of these DNA methylation changes. Hence, the conclusions are too speculative.  

Author Response

Reviewer 2:

This structured review discusses DNA methylation as a potential mechanism by which specific chemicals from WTC dust act as carcinogens. The authors searched Pubmed for studies that investigated DNA methylation changes in humans exposed to carcinogens that had been detected in individuals exposed to WTC dust. Eighty studies were included in their analysis. These studies indicated that various carcinogens altered DNA methylation at specific genomic sites. However, there is no evidence that the levels or exposures of individuals exposed to WTC dust result in any of these DNA methylation changes. Hence, the conclusions are too speculative. 

We thank reviewer 2 for taking the time to review our work. Our first objective was to review which carcinogens were present in the WTC dust, and if these carcinogens have ever been found to be elevated in the blood or tissues of WTC-exposed persons. We hope that this is a meaningful contribution to the literature, given that cancer rates are elevated for WTC responders.

Additionally, we sought to explore some potential biological avenues whereby WTC dust may be contributing towards carcinogenesis given the current literature. While our previous EWAS studies of WTC exposure have shown differential DNA methylation in important cancer-related genes and pathways, we acknowledge that no direct evidence is provided in this manuscript to validate in WTC-exposed individuals the carcinogen-associated DNA methylation changes reported in the literature. We’ve therefore added substantially to the literature, “…we’ve observed increased global DNA methylation, and statistically significant alterations in the DNA methylation levels of cancer genes and pathways, among WTC-exposed persons.138 For instance, tumor suppressor BRCA1 has previously been observed to by hypermethylated among WTC exposed persons, and it’s hypermethylation is associated with PAH exposure as reviewed here.” (Lines 264-266)

And “It should be acknowledged that this literature review only summarizes potential biological avenues whereby WTC dust may be contributing towards carcinogenesis, without providing direct evidence that exposure of individuals to WTC dust resulted in the DNA methylation change mentioned here. Additional direct evidence from larger EWAS as well as animal studies is warranted to test these hypothesized associations.” (Lines 271-275)

These sentiments are also reiterated in our conclusion, that this review should be the basis for future, validating work. Never the less we feel that this review is an important contribution to the field and generates insightful hypothesis to be evaluated in future work.

Reviewer 3 Report (New Reviewer)

Comments and Suggestions for Authors

The manuscript provides a comprehensive review on the presence of human carcinogens in World Trade Center (WTC) dust, which are also detected in biological samples from WTC-exposed individuals, and the identification of DNA methylation dysregulation as a probable mechanism leading to cancer. The authors also perform a comprehensive analysis and summary for the effect of the carcinogens on DNA methylation. The application of Covidence systematic review software to narrow down the scope of studies is interesting. This review is relevant and timely, especially considering the increasing cancer incidence among WTC-exposed populations. However, the manuscript requires revisions.

1.        The manuscript lacks clarity in its main argument, particularly in explaining why DNA methylation was selected as the primary focus over other epigenetic mechanisms, such as DNA repair or direct DNA damage. Many compounds from WTC dust affect the epigenome, but how the review justifies focusing on DNA methylation needs further explanation. A clearer rationale should be provided to distinguish why DNA methylation, specifically, is emphasized as the cause of carcinogenesis in this context. Additionally, I recommend incorporating results from your previous studies (mentioned in the Discussion, lines 259-269) that demonstrate statistically significant alterations in global DNA methylation among WTC-exposed individuals in the Introduction. Comparing these findings with DNA methylation patterns in non-WTC-exposed populations would add depth and context to your review.

2.        Comparison with non-WTC populations: It would strengthen your argument to compare the percentage of cancer cases in WTC survivors with that of the general population or non-WTC-exposed individuals (Lines 68-71). This comparison would further validate the elevated cancer risk in the WTC-exposed populations.

3.        Unify the date format throughout the manuscript. There are inconsistencies in how dates are presented (e.g., "9/11/2001", "September 11th, 2001", "9/11/01"). A consistent format will improve readability and professionalism.

4.        In Table 1, the format of the first row (black background with white text) should be changed to a white background with black text, aligning with the rest of the table.

5.        The Discussion section should explore other potential factors contributing to cancer development, such as psychological stress, lifestyle changes, or immune system alterations. These factors should be compared with DNA methylation to provide a broader understanding of cancer causality. Additionally, other epigenetic mechanisms (e.g., DNA repair and direct DNA damage) should be discussed, as these might also play a significant role in WTC-related carcinogenesis.

6.        I am curious why other abundant components of WTC dust, such as lead, mercury, and vanadium [1], which are potential carcinogens, were not mentioned in your review. Is this because they are not associated with DNA methylation and do not support your findings? If so, it would be helpful to clarify this in the manuscript.

[1] Lippmann M, Cohen MD, Chen LC. Health effects of World Trade Center (WTC) Dust: An unprecedented disaster's inadequate risk management. Crit Rev Toxicol. 2015 Jul;45(6):492-530. doi: 10.3109/10408444.2015.1044601. PMID: 26058443; PMCID: PMC4686342.

Comments on the Quality of English Language

Here are some examples that are indicative of broader issues with writing:

1.        Word choice and sentence structure: there are several awkward word choices and sentence structures. For instance, there is an overuse of "as well as" instead of "and" (e.g., line 49), making the writing sound unnatural.

2.        Wordiness: lines 55-58 are wordy and could be rewritten to improve flow and readability.

3.        Grammatical Errors: 1) Line 68: "remains" should be changed to "remain" to correct the subject-verb agreement. 2) Line 94: "conduct a structured review the literature" should be revised to "conduct a structured review of the literature" to fix the missing preposition “of”. 3) Line 123: "PAH-DNA adducts in maternal (n=170) and cord blood (n=203) was found" should be revised to "were found" to agree with the plural subject.

A thorough proofreading is necessary to address these and other potential issues in the manuscript.

Author Response

The manuscript provides a comprehensive review on the presence of human carcinogens in World Trade Center (WTC) dust, which are also detected in biological samples from WTC-exposed individuals, and the identification of DNA methylation dysregulation as a probable mechanism leading to cancer. The authors also perform a comprehensive analysis and summary for the effect of the carcinogens on DNA methylation. The application of Covidence systematic review software to narrow down the scope of studies is interesting. This review is relevant and timely, especially considering the increasing cancer incidence among WTC-exposed populations. However, the manuscript requires revisions.

  1. The manuscript lacks clarity in its main argument, particularly in explaining why DNA methylation was selected as the primary focus over other epigenetic mechanisms, such as DNA repair or direct DNA damage. Many compounds from WTC dust affect the epigenome, but how the review justifies focusing on DNA methylation needs further explanation. A clearer rationale should be provided to distinguish why DNA methylation, specifically, is emphasized as the cause of carcinogenesis in this context. Additionally, I recommend incorporating results from your previous studies (mentioned in the Discussion, lines 259-269) that demonstrate statistically significant alterations in global DNA methylation among WTC-exposed individuals in the Introduction. Comparing these findings with DNA methylation patterns in non-WTC-exposed populations would add depth and context to your review.

This is an extremely helpful point. We’ve added to our introduction, “Because 1) cancer cells are characterized by aberrant DNA methylation patterns, and 2) tumor suppressor genes, specifically, have be found to be targets to DNA hypermethyation early-on in the carcinogenic process,28 we hypothesize that DNA methylation is the form of epigenetic modification most pertinent to WTC exposure associated carcinogenesis. DNA methylation has in fact been described as “at the crossroads of gene and environment interactions.”29 Moreover, our group’s previous findings from epigenome-wide association studies of WTC exposure have shown DNA methylation changes in important cancer-related genes and pathways. For instance, enrichment in differentially methylated genes belonging to the MAPK Signaling, mTOR Signaling, Base Excision Repair, Epithelial Mesenchymal Transition, and Hypoxia pathways, among others, for WTC-exposed persons.30” (Lines 91-101)  

We have attempted to clarify our study rationale by adding to the final line of the introduction, “Here we sought to….conduct a structured review the literature on DNA methylation alterations associated with exposure to carcinogens present in the WTC dust, to identify potential biological avenues whereby WTC dust may be contributing towards carcinogenesis.”

  1. Comparison with non-WTC populations: It would strengthen your argument to compare the percentage of cancer cases in WTC survivors with that of the general population or non-WTC-exposed individuals (Lines 68-71). This comparison would further validate the elevated cancer risk in the WTC-exposed populations.

This is a great suggestion; unfortunately, we can not quantify the % or rate of cancer in WTC survivors. This is due to the nature of the World Trade Center Environmental Health Clinic (WTC EHC), which serves the healthcare needs of WTC community member survivors, primarily out of Bellevue Hospital. Patients are enrolled based on eligibility criteria defined according to their WTC exposure and having a certifiable condition or illness. Thus healthy WTC survivors are not enrolled and it impossible to quantify the denominator of WTC-exposed survivors. Statistically significant rates of excess cancer have, however, been quantified among WTC responders as stated in the introduction.

  1. Unify the date format throughout the manuscript. There are inconsistencies in how dates are presented (e.g., "9/11/2001", "September 11th, 2001", "9/11/01"). A consistent format will improve readability and professionalism.

 As suggested, all dates have been converted to the same format, September 11th, 2001.

  1. In Table 1, the format of the first row (black background with white text) should be changed to a white background with black text, aligning with the rest of the table.

 Agreed, we have made the suggested formatting changes to Table 1.

  1. The Discussion section should explore other potential factors contributing to cancer development, such as psychological stress, lifestyle changes, or immune system alterations. These factors should be compared with DNA methylation to provide a broader understanding of cancer causality. Additionally, other epigenetic mechanisms (e.g., DNA repair and direct DNA damage) should be discussed, as these might also play a significant role in WTC-related carcinogenesis.

 While we agree that these are helpful suggestions, they are, unfortunately, beyond the scope of this review. We have added to the discussion to address these limitations, “We would like to note, additionally, that this review only covers DNA methylation alterations, while carcinogens have also been linked to other types of epigenetic mechanisms, such as histone modifications.147 Moreover, beyond exposure to the dust itself, other elements of the WTC disaster event, such as the stress and trauma, may have caused epigenetic modifications.148 Future studies should explore these issues to develop a broader understanding of WTC exposure and cancer causality.” (Lines 341-346)

  1. I am curious why other abundant components of WTC dust, such as lead, mercury, and vanadium [1], which are potential carcinogens, were not mentioned in your review. Is this because they are not associated with DNA methylation and do not support your findings? If so, it would be helpful to clarify this in the manuscript.

In this review, we choose to include only IARC Group 1 carcinogenic with “sufficient evidence of carcinogenicity in humans.” Lead, for instance, is currently classified as Group 2A “probably carcinogenic to humans.” This is explained in our methods section, lines 110-111.

[1] Lippmann M, Cohen MD, Chen LC. Health effects of World Trade Center (WTC) Dust: An unprecedented disaster's inadequate risk management. Crit Rev Toxicol. 2015 Jul;45(6):492-530. doi: 10.3109/10408444.2015.1044601. PMID: 26058443; PMCID: PMC4686342.

This paper is already cited in this review, it is reference #2.

Comments on the Quality of English Language

Here are some examples that are indicative of broader issues with writing:

  1. Word choice and sentence structure: there are several awkward word choices and sentence structures. For instance, there is an overuse of "as well as" instead of "and" (e.g., line 49), making the writing sound unnatural.

 Thank you for bringing this to our attention, we’ve corrected to “and” throughout the text.

  1. Wordiness: lines 55-58 are wordy and could be rewritten to improve flow and readability.

We have re-writing this sentence as suggested. “These exposures resulted from the fires that persisted through December 2001, from dust during recovery and clean-up work, and from resuspended dust in the streets, in and incompletely or poorly cleaned indoor areas, and in buildings ventilation systems.7

  1. Grammatical Errors: 1) Line 68: "remains" should be changed to "remain" to correct the subject-verb agreement. 2) Line 94: "conduct a structured review the literature" should be revised to "conduct a structured review of the literature" to fix the missing preposition “of”. 3) Line 123: "PAH-DNA adducts in maternal (n=170) and cord blood (n=203) was found" should be revised to "were found" to agree with the plural subject.

Thank you, we’ve made these helpful corrections.

A thorough proofreading is necessary to address these and other potential issues in the manuscript.

We have proofread the manuscript as suggested and corrected grammatical errors. 

Round 2

Reviewer 1 Report (Previous Reviewer 2)

Comments and Suggestions for Authors

No more comment.

Author Response

This reviewer did not have additional comments. We thank them again for their time reviewing our work. 

Reviewer 2 Report (Previous Reviewer 1)

Comments and Suggestions for Authors

The clarifications provided in the cover letter and the additional information and discussion have adequately addressed my previous concerns.

Author Response

This reviewer did not have additional comments. We thank them again for their time reviewing our work. 

Reviewer 3 Report (New Reviewer)

Comments and Suggestions for Authors

The revision addressed most of my concerns. The authors have substantially improved the manuscript, and I commend them for addressing the key issues raised in the initial review. However, there are still a few minor adjustments that could further enhance the clarity and presentation of the paper:

  1. Improve the resolution of Figure 1
  2. Clarify DNA methylation effects for each element: the Table 2 summarizing the effects of different carcinogens on DNA methylation is informative, but it could benefit from more clarity. Specifically, it would be helpful to indicate whether each carcinogen increases or decreases the methylation of the genes listed. In the current format, it’s unclear whether the genes undergo hypermethylation or hypomethylation.

Overall, these are minor revisions that would improve the readability and comprehension of the manuscript. The content is solid, and with these small adjustments, the manuscript will be even stronger.

Author Response

Title: DNA Methylation as a Molecular Mechanism of Carcinogenesis in WTC Dust Exposure: Insights from a Structured Literature Review

Manuscript #: biomolecules-3221694

Reviewer 3:

The revision addressed most of my concerns. The authors have substantially improved the manuscript, and I commend them for addressing the key issues raised in the initial review. However, there are still a few minor adjustments that could further enhance the clarity and presentation of the paper:

Improve the resolution of Figure 1

We defer to the editorial team, and will be happy to adjust the resolution of Figure 1 if needed.

Clarify DNA methylation effects for each element: the Table 2 summarizing the effects of different carcinogens on DNA methylation is informative, but it could benefit from more clarity. Specifically, it would be helpful to indicate whether each carcinogen increases or decreases the methylation of the genes listed. In the current format, it’s unclear whether the genes undergo hypermethylation or hypomethylation.

This information is provided in Supplementary Table S2. To clarify, we’ve added “Results are described fully in Table 2 and summarized below. Additional details about each study, including direction of DNA methylation (increase or decrease) per gene for each carcinogen, are provided in Supplementary Table S2.” (Lines 1999-202)

Overall, these are minor revisions that would improve the readability and comprehension of the manuscript

We thank the reviewer for their kind comments, and for taking the time to review our work!

This manuscript is a resubmission of an earlier submission. The following is a list of the peer review reports and author responses from that submission.

Round 1

Reviewer 1 Report

Comments and Suggestions for Authors

This manuscript is very similar to a recently published paper ( PMID: 38131903) by the same group. Hence, there is not sufficiently new insight into the impact of carcinogens from the World Trade Center dust on DNA methylation to warrant an additional publication.

Reviewer 2 Report

Comments and Suggestions for Authors

Dr. Tuminello et al. evaluate the exposure of chemical carcinogens from WTC dust among responders and other survivors and performed literature studies, to shine light on the mechanism of increased risk of cancer among these subjects. They found that many human carcinogens such as Arsenic and asbestos (not from WTC dust) induced DNA methylation alterations in subjects with professional exposure to these chemicals. Thus, they claimed that WTC dust may cause DNA methylation changes that may contribute to the increase of cancer risk by WTC dust. However, the levels of DNA methylation changes in WTC collapse responders or other survivors  were not determined. It is unknown whether the exposure doses of carcinogens from WTC dust are high enough to induce DNA methylation changes. Although we could not exclude the contribution of DNA methylation mechanism to the development of cancer by WTC dust, no direct evidence has been presented in their manuscript to support the claimed mechanism.